# Charge guides pathway selection in β-sheet fibrillizing peptide co-assembly

Dillon T. Seroski[1], Xin Dong[2], Kong M. Wong [3], Renjie Liu[1], Qing Shao[2], Anant K. Paravastu[3], Carol K. Hall[2] & Gregory A. Hudalla [1✉]

Peptide co-assembly is attractive for creating biomaterials with new forms and functions. Emergence of these properties depends on the peptide content of the final assembled structure, which is difficult to predict in multicomponent systems. Here using experiments and simulations we show that charge governs content by affecting propensity for self- and co-association in binary CATCH(+/−) peptide systems. Equimolar mixtures of CATCH(2+/2−), CATCH(4+/4−), and CATCH(6+/6−) formed two-component β-sheets. Solid-state NMR suggested the cationic peptide predominated in the final assemblies. The cationic-to-anionic peptide ratio decreased with increasing charge. CATCH(2+) formed β-sheets when alone, whereas the other peptides remained unassembled. Fibrillization rate increased with peptide charge. The zwitterionic CATCH parent peptide, "Q11", assembled slowly and only at decreased simulation temperature. These results demonstrate that increasing charge draws complementary peptides together faster, favoring co-assembly, while like-charged molecules repel. We foresee these insights enabling development of co-assembled peptide biomaterials with defined content and predictable properties.

[1] J. Crayton Pruitt Family Department of Biomedical Engineering, University of Florida, Gainesville, FL 32611, USA. [2] Department of Chemical and Biomolecular Engineering, North Carolina State University, 911 Partners Way, Raleigh, NC 27695, USA. [3] School of Chemical and Biomolecular Engineering, Georgia Institute of Technology, Atlanta, GA 30332, USA. ✉email: ghudalla@bme.ufl.edu

Peptides that assemble into elongated fibrillar structures (i.e., "peptide nanofibers") are widely used to create biomaterials for medical and biotechnology applications[1–5]. Co-assembly is a process in which two distinct peptides (e.g., A and B) can associate to form a single supramolecular structure. Combining two different types of molecules into a single fibrillar architecture can lead to biomaterials with a new form and function[6–8]. Early examples relied on the co-assembly of a peptide A with a variant of A (e.g., A*) that is modified to include a molecule that imparts specific biological activity, such as cell adhesion, molecular recognition, or antigen presentation. This strategy has been widely used to create nanomaterials for tissue engineering and regenerative medicine[9–15], immune engineering[16–18], protein capture and release[19–21], and fouling resistance[22], among other applications. More recent examples combine two different peptides, A and B, that can self-associate (i.e., AA or BB) as well as co-assemble (i.e., AB). This approach can lead to biomaterials with new nanoscale structures[23], hydrogelators[24], gradated protein display[25], scaffolding for anchorage-dependent cells[26], drug delivery[27], and insights into protein-misfolding diseases[28].

Changes in peptide amino acid sequence often alter nanofiber properties, and predicting these changes requires an understanding of how molecular features guide selection among the different assembly pathways. Within single-component systems, for example, replacing positive residues with negative residues, and vice versa, can alter nanofiber morphology or assembly kinetics[29–34]. Within binary systems, the possibilities are quite complicated, as peptides can cooperatively co-assemble, randomly co-assemble, destructively co-assemble, or self-sort[6]. Prior work has shown that when two oppositely charged peptides co-assemble, the resulting nanofibers can have different morphologies than those observed after self-association of either peptide alone[28,35–40]. When peptides self-sort and form distinct nanostructures, interactions between these structures can alter bulk biomaterial properties[38]. A grand challenge in peptide co-assembly is having the ability to accurately predict biomaterial structure and function from amino acid sequence information. Unfortunately, these features are determined by peptide content of the nanofiber, which is difficult to ascertain in multicomponent systems due to the inability to distinguish self- and co-association events using conventional approaches.

"Selective co-assembly," is a unique case wherein two distinct peptides, A and B, resist self-association when alone, but when combined form fibrillar structures. Selective co-assembly is in many ways analogous to triggered self-assembly, wherein peptide association is induced by introducing a chemical (e.g., pH, salt)[41–43] or physical (e.g., heat, light)[44,45] stimulus. Often, selective co-assembly is enabled by the cooperative transition of peptides A and B from random-coil conformations into secondary structures that template the intermolecular interactions that drive fibrillization, such as α-helices[46–48] and β-strands[49–51]. Like triggered self-assembly, selective co-assembly is advantageous for encapsulating sensitive biological cargoes, such as cells[50,51] and proteins[52]. We envision that selective co-assembly can also provide control of pathway selection in binary peptide systems by minimizing the competition between self- and co-association events.

To date, most peptides that selectively co-assemble have been derived from self-assembling peptides by swapping out neutral residues for charged residues such as lysine (K), arginine (R), glutamic acid (E), and aspartic acid (D). Examples of peptide co-assemblies designed using this concept include KVW10 and EVW10 derived from MAX1[49], P$_{11}$-13 and P$_{11}$-14 derived from P$_{11}$-2[50], CATCH(+) and CATCH(−) derived from Q11[52],

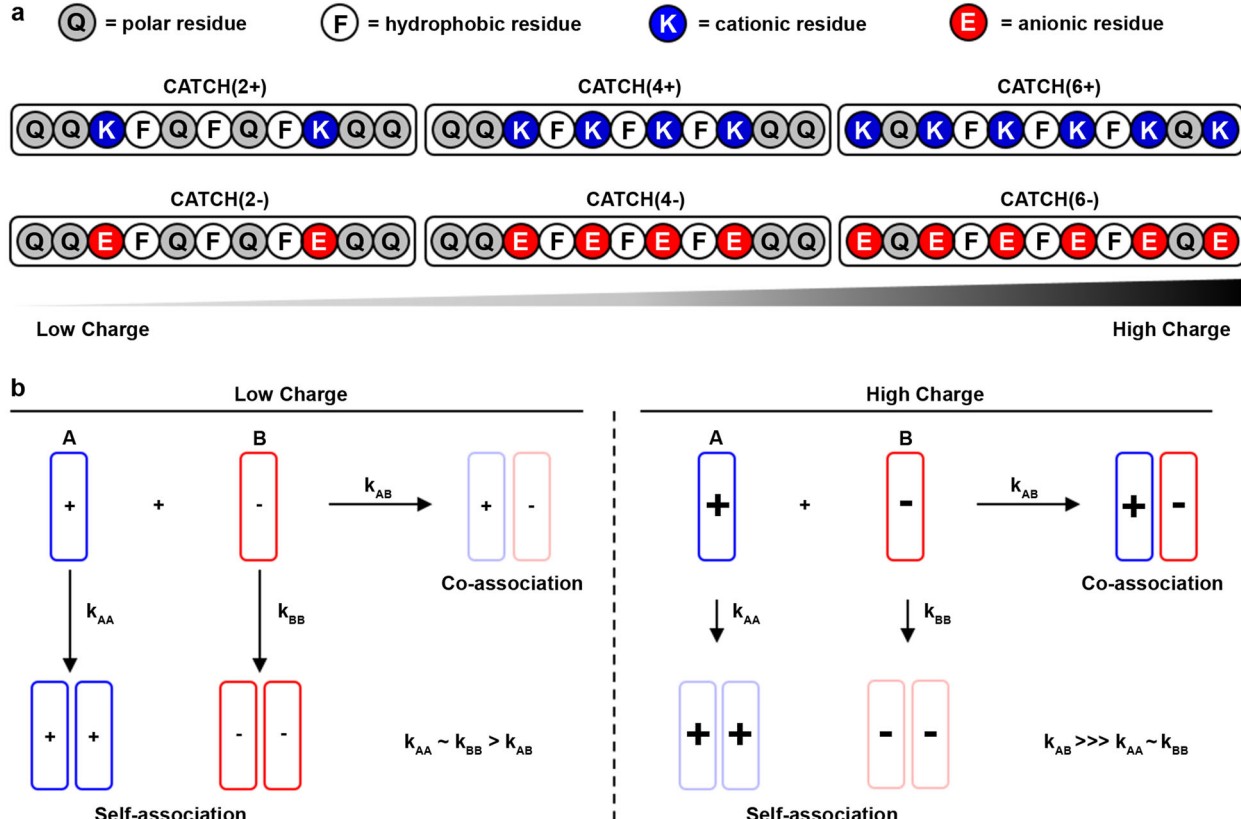

**Fig. 1 Schematic of CATCH peptides and their assembly. a** Schematic of residue placement in CATCH peptides. **b** Schematic of hypothesized reaction rates based on the number of charged residues.

and KLVFWAK and ELVFWAE derived from Aβ[53]. Creating highly charged variants of self-assembling peptides is thought to prevent their self-association through electrostatic repulsion and facilitate co-assembly through electrostatic attraction and hydrophobic collapse. We recently reported that the charge-imbalanced peptide pair CATCH(4+) and CATCH(6−) selectively co-assembles into nanofibers predominated by an alternating AB strand order, although a non-negligible number of AA and BB "mismatches" are predicted from models and observed experimentally[54]. The peptide component with the lower number of charged residues had greater propensity for strand mismatching (i.e., "self-association"). This observation may not be surprising, but when coupled with the limited sequence space that has been explored in peptide co-assembly, underscores the need for systematic studies to understand how molecular features influence nanofiber formation, composition, structure, and function.

Here, we sought to understand the role of charge in the co-assembly of binary peptide mixtures using the CATCH system. To maintain consistency with the original CATCH(4+/6−) peptides, which were derived from the self-assembling peptide Q11[55], we systematically replaced glutamine residues with either positively charged lysine, K, or negatively charged glutamic acid, E, residues. This design incrementally varies the number of charged residues in each peptide, but retains the hydrophobic core required for β-sheet fibrillization (Fig. 1a)[52]. We proposed that the number of charged amino acid residues within each peptide would determine selection between self-association and co-association pathways (Fig. 1b). Biophysical experiments and computational models identified lower limits for the number of cationic or anionic amino acids that bias CATCH peptide mixtures away from self-assembly and toward co-assembly at equilibrium. Discontinuous molecular dynamic (DMD) simulations based on the PRIME20 model, which were corroborated by experiments, identified an influence of peptide charge on co-assembly kinetics that was not anticipated based on prior knowledge. These results provide a mechanistic basis for pathway selection that extends beyond the concept of a thermodynamic "on-off" switch, and identify charge as a fundamental design consideration for controlling peptide co-assembly.

## Results

**Self-association and co-assembly of CATCH pairs.** We characterized co-assembly of the CATCH(2+/2−), CATCH(4+/4−), and CATCH(6+/6−) pairs in 1× phosphate-buffered saline (PBS) using Fourier transform infrared (FTIR) spectroscopy, transmission electron microscopy (TEM), solid-state nuclear magnetic resonance (ssNMR) spectroscopy, and DMD simulations (Fig. 2). We first used TEM to view nanofiber morphology 24 h after combining CATCH(2+/2−), CATCH(4+/4−), or CATCH(6+/6−) in solution at an equimolar ratio (Fig. 2a). CATCH(2+/2−) formed nanofibers with long persistence lengths that were prone to aggregation (Fig. 2a and Supplementary Fig. 1), similar to Q11 (Supplementary Fig. 2). In contrast, CATCH(6+/6−) formed nanofibers with uniform widths of $11.5 \pm 2.6$ nm that spanned hundreds of nanometers with very short persistence lengths and little to no lateral association (Fig. 2a and Supplementary Fig. 3). CATCH(4+/4−) formed nanofibers that were similar to both CATCH(6+/6−) and CATCH(2+/2−) with regions of short and long persistence lengths as well as regions with high and low lateral association (Fig. 2a and Supplementary Fig. 4). Similar differences in assembled nanofiber morphology were reported previously in a study comparing charged one- and two-component peptide nanofibers[56].

We used FTIR spectroscopy in the amide I region between 1600 and 1700 cm$^{-1}$ to analyze the formation of new hydrogen bonds resulting from peptide organization into structures rich in β-sheets

(Fig. 2a). Samples of the CATCH parent peptide Q11 had a strong maximum at 1622 cm$^{-1}$ indicative of a β-sheet conformation (Supplementary Fig. 5), consistent with a prior report[13]. Similarly, FTIR spectra for equimolar mixtures of CATCH(2+/2−), CATCH(4+/4−), and CATCH(6+/6−) had strong maxima between 1619 and 1620 cm$^{-1}$, indicative of β-sheet secondary structures within the samples (Fig. 2b, black traces)[57]. In contrast, spectra of CATCH(2−), CATCH(4+), CATCH(4−), CATCH (6+), and CATCH(6−) had broad maxima at ~1642–1645 cm$^{-1}$ indicative of random-coil configurations (Fig. 2b, blue and red traces). Unexpectedly, the CATCH(2+) spectrum had a maximum at 1621 cm$^{-1}$, indicating a predominance of β-sheet secondary structure (Fig. 2b, blue trace, left panel); this result suggested that CATCH(2+) can self-associate. Notably, CATCH (2+) in pure water also had a maximum at 1620 cm$^{-1}$ (Supplementary Fig. 6), suggesting that self-association does not depend on counterions in 1× PBS.

The secondary structures and content of CATCH(2+/2−), CATCH(4+/4−), and CATCH(6+/6−) samples in the gel state were assessed through 1D $^{13}$C ssNMR spectra (Fig. 2c and Supplementary Fig. 7). Samples were first subjected to centrifugation to separate assembled peptide from unassembled peptide and then lyophilized before conducting ssNMR measurements. 1D $^{13}$C spectra were collected using the composite-pulse multi-CP method from Duan and Schmidt-Rohr[58], which was validated with a series of similar co-assembled β-sheet peptide samples (Supplementary Figs. 8–13) and allows quantitative peak comparison. Narrow linewidths observed in the aliphatic region of 1D $^{13}$C spectra ranged from 1.1 to 1.8 p.p.m., consistent with linewidths observed in previous amyloid fibrils, indicating that each peptide pair assembled into an ordered structure[59,60]. The $^{13}$C chemical shifts for samples of CATCH(2+/2−), CATCH(4+/4−), and CATCH(6+/6−) were consistent within peak linewidths (Supplementary Table 1 and Supplementary Fig. 14), suggesting that these peptide pairs adopted similar structures. Secondary chemical shifts for carbonyl carbons and α-carbons, which are calculated as the change in chemical shift value from values reported for random-coil structures, can be used to determine secondary structure[61]. Here, CATCH(2+/2−), CATCH(4+/4−), and CATCH(6+/6−) showed approximate secondary chemical shifts of −2.7 and −0.7 p.p.m. for the carbonyl and α-carbon, respectively, indicating that the samples were rich in β-sheets, consistent with FTIR measurements. Further, the presence of shifts corresponding to the C$_\gamma$ of lysine and C$_\delta$ of glutamic acid at ~23 and ~181 p.p.m., respectively, indicated that both (+) and (−) peptides were present in the samples; however, the magnitude of the signal suggested that the cationic peptide was the predominant species. Notably, the much smaller peak at 179 p.p.m. in the CATCH(2+/2−) sample assigned to the C$_\delta$ of glutamic acid suggested that significantly less CATCH (2−) was present in the centrifuged pellet compared to CATCH (2+). The ratio of cationic to anionic peptide decreased with increasing charge, approaching 2 in CATCH(6+/6−) samples. Measured ratios were comparable to the estimated ratio of cationic to anionic peptide in CATCH(4+/6−) reported previously[54]. We note that the exact position of the C$_\delta$ of glutamic acid may differ slightly from sample to sample. We postulate that the different nanofiber morphologies observed in TEM micrographs in Fig. 2a may arise in part from the differences in the amount of CATCH(+) and CATCH(−) peptides detected in the different CATCH nanofiber samples with NMR.

DMD simulations with the PRIME20 force field predicted aggregation of equimolar mixtures of CATCH(4+/4−) and CATCH(6+/6−) into bilayer β-sheet structures (Fig. 2d and Supplementary Videos 1 and 2), in agreement with the experimental data; however, CATCH(2+/2−) was not predicted to aggregate (Fig. 2d) unless the simulation temperature was

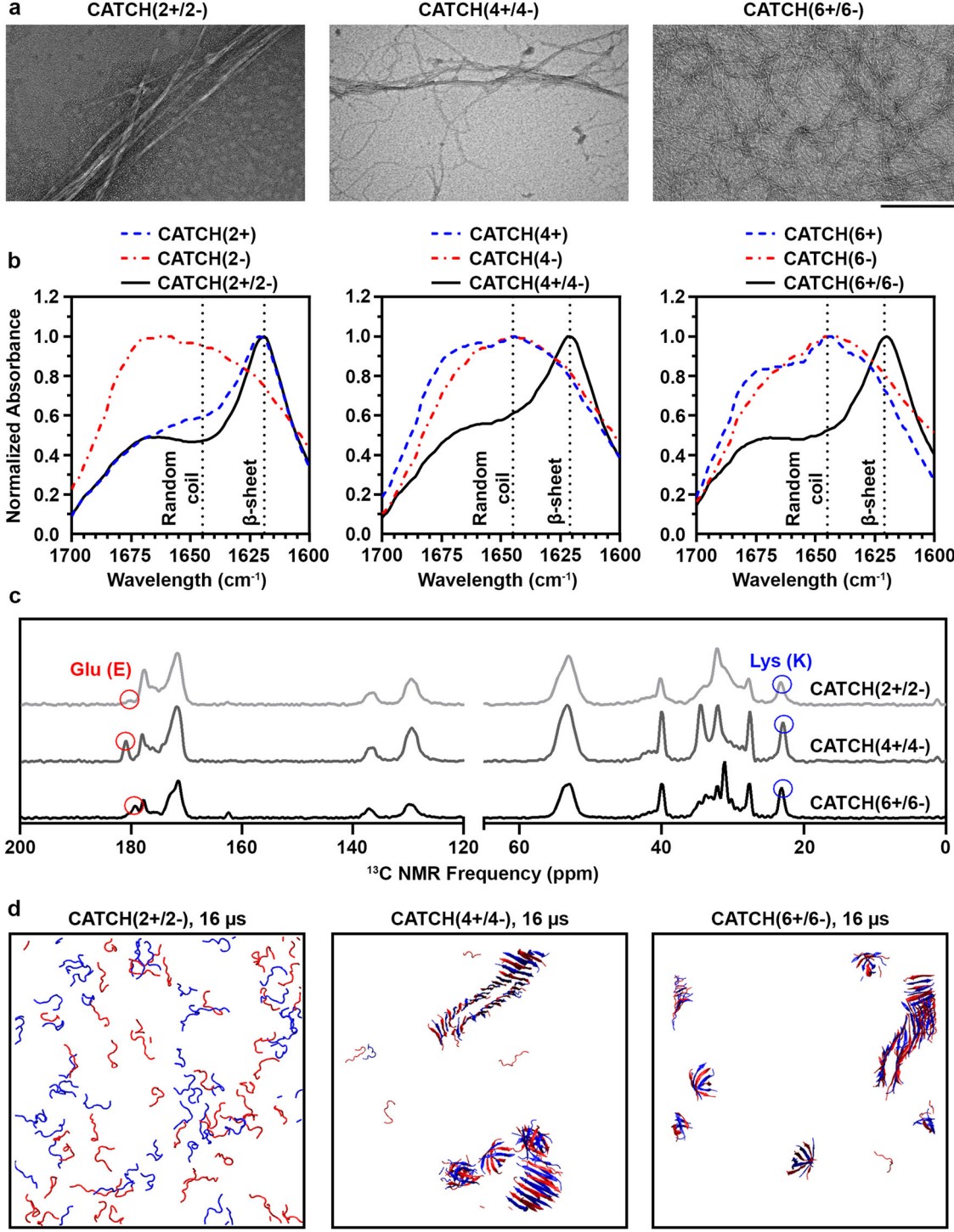

**Fig. 2 Charge-complementary CATCH peptide pairs form β-sheet nanofibers. a** TEM micrographs of CATCH(2+/2−), CATCH(4+/4−), and CATCH (6+/6−) (scale bar = 200 nm). **b** FTIR-ATR spectra of CATCH(2+) (dashed blue line), CATCH(2−) (dashed red line), CATCH(2+/2−) (solid black line), CATCH(4+) (dashed blue line), CATCH(4−) (dashed red line), CATCH(4+/4−) (solid black line), CATCH(6+) (dashed blue line), CATCH(6−) (dashed red line), CATCH(6+/6−) (solid black line). **c** CPMAS of co-assembled CATCH(2+/2−) (light gray line), CATCH(4+/4−) (dark gray line), and CATCH (6+/6−) (black line). **d** DMD simulations of co-assembled CATCH(2+/2−), CATCH(4+/4−), and CATCH(6+/6−) after 16 μs where cationic and anionic peptides are represented by blue and red, respectively.

decreased from $T^* = 0.2$ to $T^* = 0.18$ (Supplementary Fig. 15). We hypothesized that the lack of observed aggregation of CATCH(2+/2−) in DMD models at higher temperatures could be due to slower kinetics that were beyond the 16 μs time scale of the simulations. Analysis of the final aggregated structures of CATCH(4+/4−) and CATCH(6+/6−) indicated that

neighboring peptides in these co-assemblies formed hydrogen bonds and that the phenylalanine sidechains pointed inward toward the core of the bilayers, while the charged sidechains pointed outward toward (the implicit) water, similar to DMD models reported previously for equimolar mixtures of CATCH (4+) and CATCH(6−)[54]. Furthermore, the peptides within each

β-sheet were predominantly organized into an alternating positive/negative configuration (e.g., CATCH(+):CATCH(−):CATCH(+):CATCH(−)), although some CATCH(+):CATCH(+) and CATCH(−):CATCH(−) neighbors were observed. No stable aggregates were observed in DMD simulations of CATCH(2−), CATCH(4+), CATCH(4−), CATCH(6+), and CATCH(6−) alone (Supplementary Fig. 16), consistent with FTIR measurements, suggesting that the presence of a charge-complementary peptide is required for assembly of these molecules. Aggregates were also not observed in DMD simulations of CATCH(2+) alone (Supplementary Fig. 10), in contrast to experiments, suggesting that self-assembly of this molecule may proceed over longer timescales than the simulation window. Taken together, simulations and experiments demonstrate that the CATCH(6+/6−) and CATCH(4+/4−) peptide pairs selectively co-assemble into two-component β-sheet nanofibers. In contrast, CATCH(2−) may cooperatively co-assemble with CATCH(2+); however, the self-assembly propensity of CATCH(2+) favors its increased incorporation into the resultant assemblies.

**Minimum concentration required for CATCH co-assembly.** Next, we evaluated the minimum concentration required for co-assembly of CATCH(4+/4−) and CATCH(6+/6−) pairs using CD (Fig. 3 and Supplementary Fig. 17). CATCH(2+), CATCH(2−), and the pair were not studied due to the difficulty of distinguishing co- and self-assembly events. Samples of CATCH(4+), CATCH(4−), CATCH(6+), and CATCH(6−) alone all had CD spectra with minima between 199 and 202 nm indicative of random-coil configurations (Supplementary Fig. 17). Over the range of 50–400 μM, equimolar mixtures of CATCH(4+/4−) and CATCH(6+/6−) transitioned from predominantly random-coil configurations to a

mixture of random coils and β-sheets, and to predominantly β-sheet secondary structures (Fig. 3a). The minimum concentration at which CATCH(4+/4−) and CATCH(6+/6−) formed a predominantly β-sheet secondary structure was between 300 and 400 μM (Fig. 3b), which is consistent with the minimum reported previously for equimolar mixtures of CATCH(4+/6−)[52]. However, more β-sheet content, indicated by the minimum at 212 nm, was observed at lower concentrations of CATCH(6+/6-) than CATCH(4+/4−), suggesting that increasing the number of charged residues decreases the threshold for co-assembly.

**Co-assembly kinetics observed via DMD simulations and biophysical experiments.** DMD simulations offer a unique opportunity to probe co-assembly kinetics at the microsecond scale, which is not accessible using conventional biophysical experimentation. Here, we examined DMD snapshots of equimolar mixtures of CATCH(4+/4−) and CATCH(6+/6−) containing 96 peptide units over 200 billion collisions (~16 μs) at time points of 0, 1.6, 3.2, 8, and 16 μs (Fig. 4a, b and Supplementary Fig. 18). At $t = 0$ μs, the simulation randomly distributes the peptide units throughout the box in random-coil configurations. For the CATCH(4+/4−) pair, a snapshot at 1.6 μs revealed the initiation of some bilayer β-sheet secondary structures with most of the peptide units still unassembled. After 3.2 μs, we observed an increase in the number of assembled peptides, but it was not until 8 μs that approximately half of the peptides were incorporated into bilayer structures. Finally, after 16 μs, nearly all of the peptide units were incorporated into β-sheet nanofiber structures. Snapshots of CATCH(6+/6−) show an increase in the number of peptides assembled into β-sheet fibrils over time relative to CATCH(4+/4−). In particular, after 1.6 μs a fraction of CATCH(6+) and CATCH(6−) peptides were already incorporated into bilayer β-sheet secondary structures. By 3.2 μs, about half of the peptides were in the assembled state, at 8 μs most were assembled, and after 16 μs nearly all of them were incorporated into β-sheet nanofiber structures. Recall that the CATCH(2+/2−) pair did not form aggregates after 16 μs in simulations at $T^* = 0.2$ (Fig. 2d).

Treating the number of hydrogen bonds formed between peptides in the simulated environment as a measure of the co-assembly kinetics suggested that CATCH(6+/6−) fibrillization proceeded faster than that of CATCH(4+/4−) (Fig. 4c). A closer examination of the DMD assembly kinetics demonstrated a faster rate of free peptide depletion as well as faster increase in the number of peptides within fibrillar structures for CATCH(6+/6−) relative to CATCH(4+/4−) (Fig. 4d). The number of hydrogen bonds formed in the DMD simulations (Fig. 4c) correlates with the fibril formation kinetic profile (Fig. 4d), whereas the number of oligomers remains relatively constant, suggesting that free peptides preferentially add onto already established β-sheets.

Informed by the DMD simulations, CATCH peptide co-assembly kinetics were evaluated using CD spectroscopy and thioflavin T (ThT) fluorimetry (Fig. 5a, b). The CD spectra of samples of CATCH(6+/6−) were relatively constant over the range of 1–60 min, and the strong minimum in ellipticity at 212 nm suggested most of the peptides adopted a β-strand conformation within 1 min (Fig. 5a). In contrast, the magnitude of the ellipticity at 212 nm in samples of CATCH(4+/4−) increased with time, suggesting that these peptides adopted a β-strand conformation more slowly than the CATCH(6+/6−) pair of peptides. Relative to each of the peptides alone, samples of CATCH(4+/4−) and CATCH(6+/6−) exhibited a stronger fluorescence signal from ThT (Supplementary Fig. 19), a dye that demonstrates an increase in fluorescence upon binding to cross-β amyloid structures[62]. Measuring changes in ThT fluorescence over time, CATCH(6+/6−) samples reached a

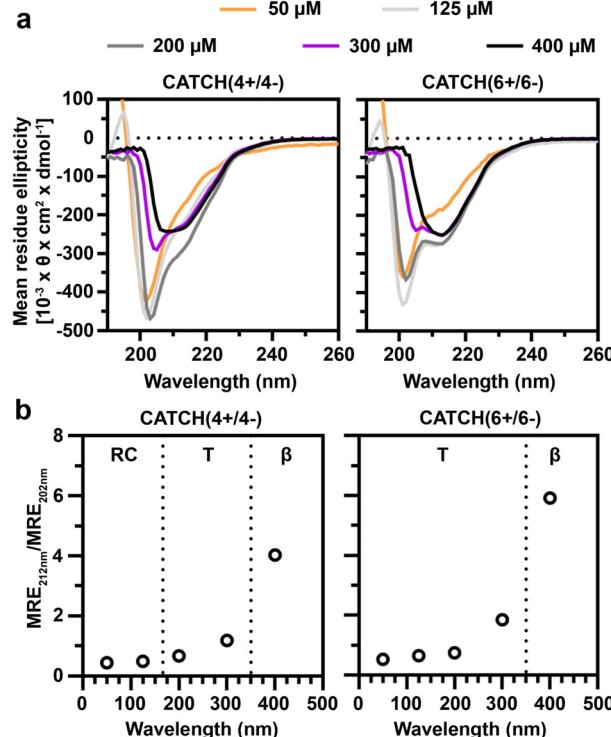

**Fig. 3 CD spectra of CATCH(4+/4−) and CATCH(6+/6−) mixtures.**
**a** Concentration-dependent transition from random coil to β-sheet of **a** CATCH(4+/4−) and CATCH(6+/6−) at 50 μM (orange line), 125 μM (light gray line), 200 μM (dark gray line), 300 μM (purple line), and 400 μM (black line). **b** Concentration-dependent CD spectra plotted as $MRE_{212 nm}/MRE_{202 nm}$. RC = random-coil region, T = random coil to β-sheet transition region, β = β-sheet region.

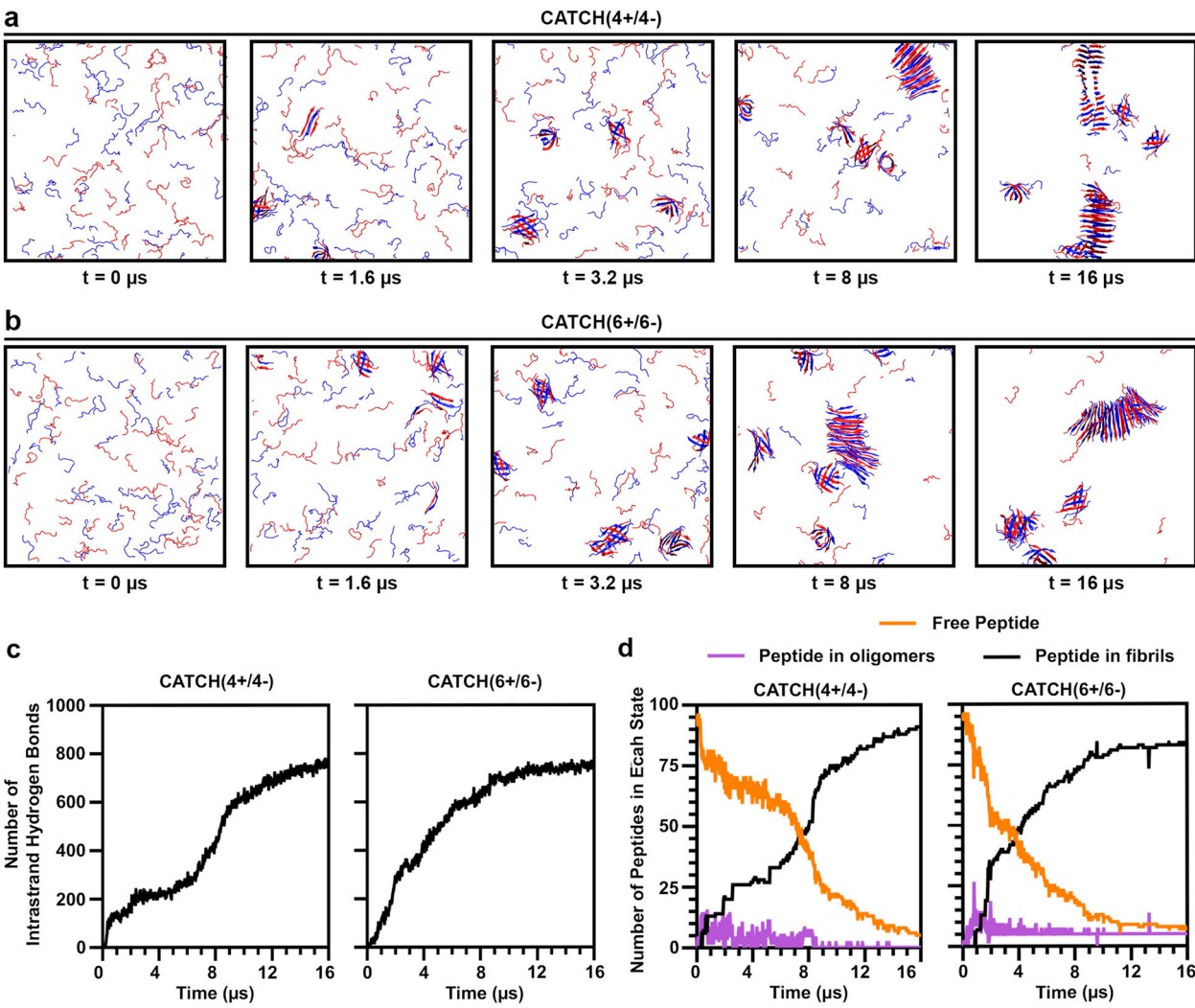

**Fig. 4 Computational analysis of CATCH co-assembly kinetics.** DMD snapshots of **a** CATCH(4+/4−) and **b** CATCH(6+/6−) over time where cationic and anionic peptides are represented by blue and red, respectively. **c** Quantitative assessment of hydrogen bond formation over time. **d** DMD analysis of the free peptide depletion (solid orange line), oligomerization (association of 2–5 peptides, solid purple line), and fibrillization (addition of free peptides onto assemblies with >5 peptides, solid black line) of CATCH(4+/4−) and CATCH(6+/6−) over time.

maximum between 12 and 24 h, while CATCH(4+/4−) samples required 24–48 h (Fig. 5a). Notably, the faster kinetics observed through ThT fluorescence suggested that CATCH(6+/6−) may have a faster rate of fibril formation than CATCH(4+/4−), as longer nanofibers would provide more sites for ThT to bind[63]. The co-assembly kinetics of the CATCH(4+/4−) and CATCH (6+/6−) peptides depended on the ionic strength of the solution, whereas self-assembly of the individual peptides generally did not (Supplementary Figs. 20–25). In particular, the time to half-max of the ThT signal was shorter for mixtures of CATCH(4+/4−) in 5× PBS when compared to 1× PBS; however, the time to half-max ThT signal was longer for samples of CATCH(4+/4−) in 10× PBS (Supplementary Fig 20). In contrast, the time to half-max for mixtures of CATCH(6+/6−) decreased with increasing ionic strength, to the extent that co-assembly kinetics could not be measured using ThT fluorimetry for CATCH(6+/6−) mixtures in 10× PBS (Supplementary Fig. 21). The type of ions present did not affect co-assembly of either CATCH(4+/4−) or CATCH (6+/6−). Comparatively, the self-assembly kinetics of zwitterionic Q11 as measured with ThT were more sluggish than either CATCH peptide pair, taking ~144 h to reach a maximum (Supplementary Fig. 26). Consistent with this, Q11 did not

aggregate in simulations at $T^* = 0.2$, but did at $T^* = 0.18$ (Supplementary Fig. 27), similar to the CATCH(2+/2−) pair.

The fibrillization process was qualitatively characterized from transmission electron micrographs (Fig. 5c). Samples of CATCH (4+/4−) at 10 μM incubated for 1 min had no visible nanofibers, but instead were predominated by non-fibrillar aggregates. In contrast, samples of CATCH(6+/6−) at 10 μM incubated for 1 min had a combination of nanofibers and non-fibrillar aggregates throughout the viewing area. At 10 min, samples of CATCH(4+/4−) had a combination of nanofibers and non-fibrillar aggregates, whereas samples of CATCH(6+/6−) were predominated by nanofibers. At 60 min, nanofibers predominated in samples of both CATCH(4+/4−) and CATCH(6+/6−). Similar observations were reported previously for equimolar mixtures of CATCH(4+) and CATCH(6−), which adopt non-fibrillar aggregates at early time points, but elongated nanofibers later[54].

## Discussion

Our results suggest that a minimum charge density is required for CATCH peptides to resist self-association. Above this minimum,

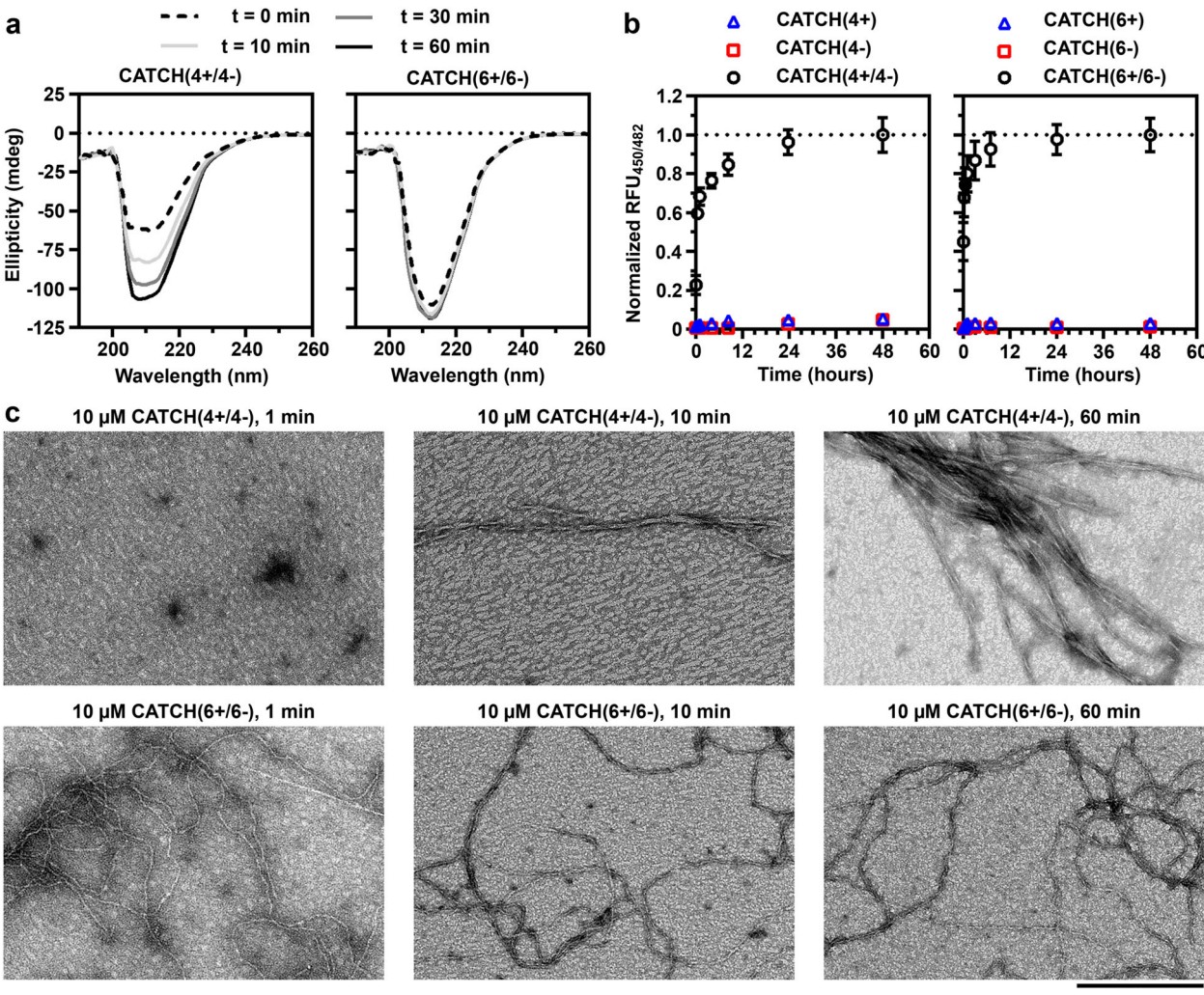

**Fig. 5 Experimental analysis of CATCH co-assembly kinetics. a** CD kinetic spectra of CATCH(4+/4−) and CATCH(6+/6−) co-assemblies over time at 400 μM ($t = 0$, dashed black line; $t = 10$ min, solid light gray line; $t = 30$ min, solid dark gray line; $t = 60$ min, solid black line). **b** Thioflavin T kinetic measurements of CATCH(4+) (blue triangles), CATCH(4−) (red squares), CATCH(4+/4−) (black circles), CATCH(6+) (blue triangles), CATCH(6−) (red squares), and CATCH(6+/6−) (black circles) at 500 μM, error bars shown as the standard error of the mean. **c** TEM of 10 μM CATCH(4+/4−) and CATCH(6+/6−) at 1, 10, and 60 min (scale bar = 200 nm).

co-assembly kinetics are determined by the number of charged residues within the 11 amino acid sequence. Fibrillizing peptides typically require between 5 and 15 amino acids for assembly into β-sheet nanofiber architectures. Peptides too short in length do not contain enough molecular contacts to form stable hydrogen bonds, whereas longer peptides have too many degrees of freedom to form stable structures. This restriction in sequence length provides some limitations for future sequence designs, particularly with regard to the number of charged residues that can be inserted into a peptide while maintaining a sufficient number of hydrophobic residues to favor β-sheet formation. Generally, zwitterionic self-assembling peptides with alternating hydrophilic and hydrophobic residues (e.g., Q11, RADA16, KFE8, P11-4) assemble through the expulsion of water and collapse of the hydrophobic face as well as ionic pairing, van der Waals interactions, and hydrogen bonding through the peptide backbone. The molecular interactions that drive β-sheet formation are short range (~5 nm or less) and require intimate, often directional, contacts[64,65]. In contrast, electrostatic interactions are longer range (~1 nm to 1 μm) and omnidirectional[66]. Highly charged variants of self-assembling peptides, such as CATCH(4+), CATCH(4−), CATCH(6+), and CATCH(6−) derived from Q11,

tend to resist self-association because longer-range electrostatic repulsion prevents the shorter-range interactions that drive aggregation. In contrast, peptide variants with a lesser number of charged residues, such as CATCH(2+), are prone to self-association, likely because they lack a sufficient repulsive electrostatic field strength to prevent peptide–peptide contact. A difference in preference for self- versus co-association resulted in β-sheet nanofibers with disproportionate peptide content, and correlated with observed variances in morphology, features that cannot be presently predicted from primary sequence alone.

Although co-assembly of charge-complementary peptide pairs is not a new concept, the degree of charge necessary to resist self-association has not been explicitly studied. A previous study reported that the zwitterionic enantiomeric peptides L- and D-(FKFE)₂ can co-assemble when combined, yet they also self-associate when alone[67]. A pair of peptides having +1 or −1 charged residues, EEFKWKFKEE (p1) and KKEFEWEFKK (p2), co-assemble when combined yet also self-associate at high and low pH or upon extended incubation at high ionic strength, where the former alters the charge state via acceptance or donation of protons and the latter likely mediates charge shielding via counterions present in solution[51,68]. Previous work

showed that peptide pairs with more charged residues, such as P$_{11}$-13 (EQEFEWEFEQE, charge = −6) and P$_{11}$-14 (QQOFO-WOFOQQ, charge = +4)[50], or CATCH(4+) and CATCH(6−)[52], co-assemble when combined but do not self-associate. The hydrophobic phenylalanine residues in the CATCH peptides have been shown to be necessary for co-assembly, demonstrating that molecular association is not due to charge complementarity alone[52]. Further, the ionic strength of the solution does not affect the self-assembly propensity of CATCH(4+), CATCH(4−), CATCH(6+), or CATCH(6−), except in the unique case when the ions are depleted or enriched, which favors CATCH(4+) self-association through a mechanism that is not yet clear (Supplementary Figs. 22–25). Together, these observations suggest that in aqueous solutions containing ions, more than two glutamate residues can prevent self-association, whereas three or more lysine residues are needed. These differences may reflect disparities in sidechain interaction potentials;[68] however, the effect of charge likely also depends on peptide length, number of hydrophobic residues, and the density of charged residues along the peptide backbone.

An unexpected result of our study was the dependence of co-assembly kinetics on the number of charged residues within the different CATCH peptides. Notably, both simulations and biophysical experiments predicted similar trends with regard to the kinetics of charge-complementary peptide co-assembly, despite differences in time scale and number of molecules within each system, highlighting the significance of this approach to refine our understanding of peptide fibrillization. In particular, DMD simulations, ThT fluorimetry, CD, and TEM collectively demonstrated that binary equimolar mixtures of CATCH(6+/6−) co-assemble faster than binary equimolar mixtures of CATCH(4+/4−). DMD simulations and CD suggested that CATCH(6+/6−) peptides in a equimolar mixture transition from random-coil conformations to β-strands faster than CATCH(4+/4) peptides in an equimolar mixture. ThT fluorimetry suggested that CATCH(6+/6−) co-assemblies underwent faster fibril formation than CATCH(4+/4−) co-assemblies. Solution ionic strength affected co-assembly kinetics of CATCH(4+/4−) or CATCH(6+/6−) to different extents; we observed a direct correlation between ionic strength and CATCH(6+/6−) kinetics, whereas CATCH(4+/4−) co-assembly approached a maximum at intermediate ionic strength. Further, ThT fluorimetry measurements suggested that physiologic ionic strength (i.e., 1× PBS and 1× CD buffer) may lead to increased fibril mass, estimated from the maximum ThT signal during CATCH(4+/4−) co-assembly, whereas fibril mass was not affected by ionic strength during CATCH(6+/6−) co-assembly. Although increasing ionic strength did not generally increase the self-association propensity for any of the peptides when alone, it remains to be determined if the increased rate of co-assembly at higher ionic strengths correlates with increasing content of like-charged neighbor mismatches in CATCH nanofibers.

Notably, neither CATCH(4+/4−) nor CATCH(6+/6−) demonstrated the classical "lag phase" often observed during fibril formation for peptides/proteins, such as amyloid-β and α-synuclein, which suggests that the nucleation and elongation rate constants for synthetic CATCH peptides are significantly greater than those of natural fibril-forming peptides. One plausible explanation for this is that charge serves to draw peptides together, establishing sufficiently high local concentrations of peptide to favor rapid nucleation and elongation[69]. According to Coulomb's law, attractive and repulsive forces are proportional to the number of charged residues within the peptide sequence. Assuming that the peptides are present at an exact equimolar ratio and that all lysine and glutamic acid residues are protonated or deprotonated, respectively, there is a 2.25-fold increase in Coulombic attraction between CATCH(6+) and CATCH(6−)

versus CATCH(4+) and CATCH(4−). From our observations, we postulate that the attraction between oppositely charged molecules increases their likelihood of collision, while repulsion keeps like-charged molecules apart, which together favor co-assembly over self-association. Considering the relatively slow kinetics of zwitterionic Q11, our data further support the inference that oppositely charged peptides can "attract" a partner peptide from greater distances to enable faster β-sheet assembly. When coupled with observations from DMD simulations, which demonstrated that nuclei form quickly and remain constant in number over the course of co-assembly, while free peptides preferentially add onto the ends of growing fibrils, our observations also suggest that primary nucleation plus elongation may be the dominant mechanism for CATCH peptide co-assembly. Toward this end, TEM suggested that CATCH(4+/4−) mixtures formed non-fibrillar aggregates that persisted for a longer duration than those formed in CATCH(6+/6−) mixtures. Although it remains to be determined if CATCH peptides co-assemble via a two-step nucleation mechanism involving formation of non-fibrillar aggregates that seed fibril formation[70], or if this aggregation event is off-pathway, the formation of non-fibrillar aggregates in binary CATCH(+/−) peptide mixtures suggests that the co-assembly pathway is more complicated than the single-step processes outlined in Fig. 1.

Our previous study on the CATCH(4+/6−) pair revealed the presence of like-charged neighbor "imperfections" within co-assembled nanofibers[54]. Aggregation kinetics likely play a key role in the emergence of these imperfections. We consider a co-assembling peptide system as having three possible reactions, A joins with A, B joins with B, or A joins with B, and three corresponding reaction rates (i.e., $k_{AA}$, $k_{BB}$, and $k_{AB}$). To prevent self-sorting, the rate constant of co-association needs to be significantly greater than that of self-association (i.e., $k_{AB} \gg k_{AA} \approx k_{BB}$). Further, we postulate that the formation of co-assembled peptide nanofibers lacking like-charged neighbor imperfections (i.e., perfectly alternating strand organization) requires $k_{AB}$ to be maximized, while $k_{AA}$ and $k_{BB}$ are minimized. Much like the CATCH(4+)/CATCH(6−) pair reported previously, the CATCH peptide pairs studied here may not meet these criteria, as we qualitatively observed mismatching via DMD simulations for both the CATCH(4+/4−) and CATCH(6+/6−) pairs. These observations suggest that charge plays a more significant role at long rather than short intermolecular distances in peptide co-assembly processes. Thus, charge is likely insufficient for encoding the assembly of perfectly alternating two-component nanofibers within the established sequence space. Future efforts to balance long-range charge effects with short-range intermolecular complementarity may enable opportunities to improve the compositional precision of co-assembled peptide nanofibers. We envision that achieving molecular-level precision of β-strand order within co-assembled peptide nanofibers will enable the development of new biomaterials with more sophisticated functional and structural features.

## Conclusion

In conclusion, we surveyed the assembly pathway of three pairs of charge-matched CATCH(+/−) peptide pairs, CATCH(2+/2−), CATCH(4+/4−), and CATCH(6+/6−), derived from the original charge-mismatched CATCH(4+/6−) pair. β-Sheet nanofibers were observed in all CATCH peptide mixtures, although the morphology of the nanofibers formed varied based on the number of charged amino acid residues in each peptide. ssNMR suggested that these morphological differences may arise in part from disproportionate composition of cationic and anionic peptides within the β-sheets, although other molecular determinants likely also contribute to the observed differences. We also found

that increasing charge biases binary systems toward co-assembly and away from self-assembly. These results are broadly consistent with the small sequence space of charge-complementary peptides that has been studied thus far, yet advances our understanding of specific molecular features that guide selection among the different assembly pathways that are possible. In particular, the minimum number of residues that imparts resistance to self-assembly differed between cationic and anionic peptides. Where two glutamic acid residues afforded resistance to self-association, two lysine residues did not. Further, increasing the number of charged amino acid residues within each peptide increased the rate of co-assembly of CATCH(+/−) pairs. Thus, charge can promote co-assembly in binary peptide mixtures by enhancing the rate of co-association in addition to providing a thermodynamic "on-off" switch.

## Materials and methods

**Peptide synthesis and purification**. Peptides were synthesized on a 200–400 mesh rink amide AM resin (Novobiochem) using standard Fmoc solid-phase peptide synthesis on a CS336X automated peptide synthesizer (CS Bio), according to established methods[52]. Peptides were acetylated at their N-termini with an acetylation cocktail (10% acetic anhydride (Sigma), 80% dimethylformamide (Fisher), and 10% N,N-Diisopropylethylamine (Fisher)). Synthesis resin was collected and washed with acetone (Fisher) and placed in desiccator overnight. Peptides were cleaved from resin and deprotected using a cocktail containing 95% trifluoroacetic acid (TFA) (Fisher), 2.5% triisopropylsilane (Sigma), and 2.5% ultrapure water. Soluble peptide is then separated from the solid-resin support and then precipitated using diethyl ether (Fisher) on ice for 5 min. To remove residual TFA, precipitated peptide was then pelleted via centrifugation and resuspended with cold diethyl ether three times and then dried in vacuo overnight. Peptides were dissolved in water, frozen, and freeze-dried with a FreeZone 1 lyophilizer (Labconco).

Peptides were purified to >90% purity by reverse phase high-performance liquid chromatography (RP-HPLC) using a DionexTM Ultimate 3000TM System (Thermo Scientific) equipped with a C-18 column (Thermo Scientific) or a PFP column (Thermo Scientific). The mobile phase consisted of (A) water and (B) acetonitrile, both containing 0.1% TFA. Peptides were detected by absorbance at 215 nm.

**Matrix-assisted laser desorption/ionization-time-of-flight mass spectrometry (MALDI-TOF-MS)**. For MALDI-TOF-MS analysis, RP-HPLC-purified peptide was mixed 1:1 (v/v) with α-cyano-4-hydroxycinnamic acid (Sigma) (10 mg/ml) in 70% acetonitrile and 30% water both containing 0.1% TFA. Two microliters of the solution was spotted and dried onto an MSP 96-target polished steel BC MALDI plate. Samples were analyzed using reflectron, positive mode on an AB SCIEX TOF/TOFTM 5800 (Bruker) equipped with a 1 kHz N₂ Opti-BeamTM on axis laser (Supplementary Fig. 28).

**Nanofiber preparation**. Peptides were prepared by dissolving lyophilized peptides in either water (cationic peptides) or 200 mM ammonium bicarbonate (anionic peptides) with the concentration determined using phenylalanine absorbance ($\lambda$ = 258). For all two-peptide component samples, the peptides are mixed at a 1:1 molar ratio (equimolar) in water, and are then infused with 10× PBS to reach a final concentration of 1× PBS (137 mM NaCl, 2.7 mM KCl, 10 mM Na₂HPO₄, and 1.8 mM KH₂PO₄, pH 7.4). Single-component peptide systems are dissolved and infused with 10× PBS to reach 1× PBS at the desired concentration.

**Transmission electron microscopy**. Nanofibers prepared at 1 mM were incubated overnight in 1× PBS, unless otherwise stated. Samples were diluted to 250 μM with ultrapure water filtered through a 0.22 μM syringe filter. Formvar-carbon-coated 400 mesh copper grids (FCF400-CU-UB, EMS) were floated for 30 s on top of 20 μl peptide nanofibers and then dried by tilting the grid on a Kimwipe. Samples were negatively stained with 2% uranyl acetate in water for 30 s and analyzed using a Hitachi H-700 for endpoint studies or a FEI Tecnai Spirit (FEI, The Netherlands) housed in the University of Florida Interdisciplinary Center for Biotechnology Research. TEM fiber widths were analyzed from three separate images using Image J (NIH) with the average and standard deviation reported.

**FTIR spectroscopy**. The FTIR spectra were recorded using a universal ATR sampling accessory on a Frontier FTIR spectrophotometer (PerkinElmer). Prior to scanning, the FTIR spectrophotometer was blanked with ultrapure water. Samples were prepared at 10 mM and 1× PBS with 4 μl spotted onto the ATR accessory. Each sample was scanned 50 times with the average of the spectra reported.

**Circular dichroism**. Circular dichroism was performed on a Chiroscan spectrophotometer (Applied Physics) between 190 and 260 nm. Peptides were prepared in 1× CD buffer (137 mM KF, 2.7 mM KCl, 10 mM Na₂HPO₄, and 1.8 mM KH₂PO₄, pH 7.4) at either 200 μM for single peptide solutions or 400 μM total peptide (200 μM CATCH(+) + 200 μM CATCH(−)) for co-assembled solutions, unless otherwise stated. Due to differences in assembly kinetics, samples were scanned until no changes in ellipticity were observed over time. Each sample was scanned three times with the average of the spectra reported. Data sets of peptides at different concentrations were converted to mean residue ellipticity using

$$\frac{\text{Ellipticity}}{\text{Amide bonds} \times \text{concentration} \times 10^{-7}} \times 10^{-3}.$$

**Solid-state NMR**. NMR samples at 10 mM peptide concentration in 1× PBS were recovered by centrifugation at $12{,}100 \times g$ for 5 min after 24 h of peptide assembly. Samples were freeze-dried, packed into NMR rotors, and minimally hydrated (1 mg of water per mg of peptide). Measurements were performed on an 11.75 T Bruker Avance III spectrometer with a 3.2 mm Bruker MAS probe. Quantitative $^1$H-$^{13}$C CPMAS (cross-polarization magic angle spinning) measurements were conducted at 277 K using the composite-pulse multi-CP pulse sequence[58] to compensate for effects of motion and relaxation. Samples were spun at 22 kHz to prevent spectral overlap from spinning sidebands. Measurements were run with 14 100 kHz decoupling, 0.2 s $^1$H recovery time, 2 s recycle delay, and 14 100 μs CP periods to ensure equivalent cross-polarization. Reported chemical shifts are relative to tetramethyl silane, as calibrated with adamantine before each experiment. Custom code in Wolfram Mathematica was used for chemical shift peak analysis.

**ThT kinetic analysis**. A stock solution containing 0.8 mg/ml ThT (Acros) in 10× PBS was filtered through a 0.22 μm syringe filter (Millex). Peptide samples were prepared in a black 96-well plate (Corning) to reach a final concentration of 500 μM total peptide, 0.08 mg/ml ThT, and 1× PBS. Samples were analyzed using a SpectraMax M3 spectrophotometer (Molecular Devices) using excitation 450 nm and emission 482 nm. All samples were run in triplicate, with the mean and standard deviation reported.

**Coarse-grained DMD simulations**. DMD, a fast alternative to traditional molecular dynamics, was used in conjunction with the PRIME20 force field to simulate co-assembly of CATCH peptides[71–73]. PRIME20 is an implicit solvent intermediate-resolution protein force field designed by the Hall group for peptide aggregation. In the PRIME20 model, each of the 20 natural amino acids is represented by four beads—three for the backbone spheres NH, Cα, and CO and one for the sidechain sphere R. Each sidechain sphere has a distinct hard sphere diameter (effective van der Waals radius) and sidechain-to-backbone distances (R-NH, R-Cα, and R-CO). Sidechain–sidechain interactions are modeled with a square well potential. A combination of 210 different square well widths and 19 different square well depths are used to discriminate the polar, charge–charge, and hydrophobic interactions that occur between two sidechain groups[71]. Hydrogen bonding interactions between backbone NH and CO beads are modeled as a directional square well potential, while all other non-bonded interactions are modeled as hard sphere potentials. A detailed description of the geometric and energetic parameters of the PRIME20 model is provided in earlier work[71,74,75].

Each system contained a total of 96 peptides—48 of the positively charged peptides and 48 of the negatively charged counterparts—randomly distributed in a cubic box with a side length of 200 Å. All simulations were carried out for ~16 μs in the canonical ensemble at a peptide concentration of 20 mM. The Andersen thermostat was employed to maintain the simulation at a constant reduced temperature $T^*$ of 0.20, corresponding to 342 K[76,77]. The reduced temperature in our system is defined as $T^* = kBT/\varepsilon HB$, where $\varepsilon HB$ = 12.47 kJ/mol is the hydrogen bonding energy[76].

Cluster analysis was performed to determine the rate of oligomerization and fibril formation[54]. A cluster is defined as a network of peptides that are joined through a combination of hydrophobic and hydrogen bonding interactions. A pair of peptides were considered joined if one of two conditions were met: (1) a majority of the backbone hydrogen bonding sites were occupied or (2) there is at least one hydrophobic interaction. For our system of CATCH peptides, these conditions were considered sufficient to accurately track the formation of oligomers and fibril formation over the course of a simulation.

## Data availability

Data not shown can be found in the Supplemental Materials or can be requested from the corresponding author. Initial and final trajectories for CATCH(2+/2−), CATCH(4+/4−), and CATCH(6+/6−) DMD simulations are provided in Supplementary Data 1–6.

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

## Acknowledgements

This research was supported by the National Science Foundation (RAISE 1743432). We would also like to thank Dr. Michael Harris (Dept. of Chemistry, University of Florida) for access and time on his CD spectrophotometer.

## Author contributions

D.T.S. designed, conducted experiments, analyzed data, and wrote the paper. X.D. conducted computational simulations, analyzed data, and wrote the paper. K.M.W conducted ssNMR, analyzed data, and wrote the paper. R.L. conducted TEM and edited the paper. Q.S. conducted computational simulations. A.K.P. and C.K.H. directed work in ssNMR and computational simulations, respectively, and edited the paper. G.A.H. analyzed data, directed the work, and wrote the paper.

## Competing interests

The authors declare no competing interests.
