## [Peer Review File · Communications Chemistry]

Reviewers' comments:

Reviewer #1 (Remarks to the Author):

This is an interesting paper that characterizes the fibril formation processes of a series of oppositely charged synthetic peptides, CATCH(n^+/n^-) with $n = 2, 4,$ and 6 . The authors report n -dependent differences in fibril morphology and fibrillation kinetics. These differences are replicated to some extent in DMD simulations.

This paper is a valuable contribution to the fibril formation field, especially for fibrils that are formed by designed peptides, with possible technological or biotechnological applications. The authors should address the following points:

1. Statements regarding the ratio of positively and negatively charged peptides within the fibrils, based on relative intensities of Glu Cdelta and Lys Cgamma in solid state NMR spectra, are questionable. The Glu Cdelta signal may be attenuated by effects of molecular motions on ^1H - ^{13}C dipole-dipole couplings or spin relaxation times. The authors would have to investigate the temperature dependence of the solid state NMR spectra, record cross-polarization build-up curves, and measure ^1H and ^{13}C $T_{1\rho}$ values to substantiate these statements.
2. The absence of a "lag phase" in the ThT fluorescence build-up curves is remarkable. The authors should discuss this feature and how it may relate to the fibrillation mechanism.
3. I would expect ionic strength to affect the kinetics and possibly the fibril morphologies observed for these peptides. The authors should test whether higher or lower ionic strengths have significant effects.

Reviewer #2 (Remarks to the Author):

Overall, this de novo design of beta-structure-forming peptides is highly innovative and interesting, whose sequence, self-assembling structure, and ionic-binding activity of these peptides are also important for beta-sheet-forming diseases and self-assembling nanomaterials. The authors have presented significant computational data in both main text and Supplementary Information to demonstrate their designs. The paper can be published in its present form by including some discussion about how sequence-length affects the beta-sheet forming capacity and subsequent design, any suggestion?

Reviewer #3 (Remarks to the Author):

This paper by Hudalla and colleagues is a fantastic addition to their very interesting body of work on "selective co-assembly" of peptides. The authors should be commended for the thorough nature of their approach, the nice blend of experimental and computational studies, and some creative molecular design. This paper will offer the peptide self-assembly community many new ideas in terms of facilitating systems specifically designed to co-assemble, and I look forward to the certain horizon for use of this approach to create bioactive nanostructures.

A few thoughts which arise from curiosity rather than criticism:

- 1- It is presumed the peptides are C-terminal amides, but this was not explicitly stated nor was the resin used listed in order to make this conclusion.
- 2- Is the observation for a lack of self-association reliant on a symmetric nature of the peptides? For instance, would this general peptide approach with charge conserved work for a sequence

where half the residues were D-amino acids (e.g., KQKFKFkfkqk) or the alternating structure was disrupted (e.g., KQKFKFGKFKQK)? Basically, is it just charge, or does the fit of the interface between the two molecules also play a role?

3- The authors note a large role for electrostatic repulsion in preventing self-association in the single peptide systems. This is very reasonable. Given the studies were done in PBS, the Debye length is somewhere in the <1 nm range. One wonders whether electrostatic repulsion could be overcome if the Debye length could be further shortened by even higher osmolarity (e.g., 10x PBS).

We would like to thank the reviewers for providing a comprehensive and careful review of our manuscript. Through addressing the reviewers' concerns, we feel that the manuscript has been significantly strengthened from its initial version.

In particular, we have now included new supplemental figures (Supplemental Fig. 8-13, 22-27). In addition to new data, we have made changes and clarifications to the text in accordance with reviewer suggestions. Specific changes as they relate to reviewer critiques are discussed on a point-by-point basis below. All changes in the revised manuscript are denoted by yellow highlighting.

Reviewer #1

This is an interesting paper that characterizes the fibril formation processes of a series of oppositely charged synthetic peptides, CATCH(n^+/n^-) with $n = 2, 4,$ and 6 . The authors report n -dependent differences in fibril morphology and fibrillation kinetics. These differences are replicated to some extent in DMD simulations.

This paper is a valuable contribution to the fibril formation field, especially for fibrils that are formed by designed peptides, with possible technological or biotechnological applications. The authors should address the following points:

1. Statements regarding the ratio of positively and negatively charged peptides within the fibrils, based on relative intensities of Glu Cdelta and Lys Cgamma in solid state NMR spectra, are questionable. The Glu Cdelta signal may be attenuated by effects of molecular motions on $1H$ - $13C$ dipole-dipole couplings or spin relaxation times. The authors would have to investigate the temperature dependence of the solid state NMR spectra, record cross-polarization build-up curves, and measure $1H$ and $13C$ T1rho values to substantiate these statements.

We want to thank the reviewer for pointing out possible discrepancies in our $1D$ ^{13}C NMR measurements and analysis. We agree that measurement for CP build-up and T1rho would be important to evaluate to determine whether standard CP spectra could be interpreted quantitatively. However, the composite-pulse multiCP method we used was designed to compensate for effects of motion and relaxation. We therefore used other approaches to validate these experiments. In implementing Duan and Schmidt-Rohr's composite-pulse multiCP pulse sequence⁶⁹, we measured $1D$ spectra for a uniformly ^{13}C and ^{15}N labelled sample of similar co-assembled β -sheet system, King-Webb. Measurements were conducted at 277K with a 0.2-s $1H$ recovery time, 2-s recycle delay, 10-ms acquisition time, and 0.1-ms CP ramps to match parameters used for Boc-Alanine by Duan and Schmidt-Rohr. As shown in Supplementary Fig. 8, we observed no dependence on number of CP blocks by 14 CP blocks. Comparison of fitted peak areas shown in Supplementary Fig. 9 suggest equal polarization across the range of chemical shifts consistent with Duan and Schmidt-Rohr's analysis of off-resonance irradiation. We further validated our quantitative CP measurements against ^{13}C direct pulse (DP) measurements with a 42 s recycle delay produced from 1 day of scanning. Two co-assembled β -sheet peptide nanofiber samples were tested; CATCH6K6E (the same as the CATCH66 pair reported in this manuscript) and CATCH6R6D. Supplementary Fig. 10 and Supplementary Fig. 12 show ^{13}C spectra comparing the quantitative CP and DP measurements for both CATCH6K6E and CATCH6R6D samples. No significant differences in peak intensity are

observed except for the aromatic peak at 130 ppm. Duan and Schmidt-Rohr previously noted that cross-polarization to aromatic carbons can be exceptionally challenging due to their relatively large distance to ^1H spins. We also compare the fitted peak area between the quantitative CP and DP spectra across the frequency range in Supplementary Fig. 11 and 13. Peak areas mostly agree with slight differences that may arise from variation in ^1H magnetization decay for protonated and nonprotonated carbons. In the CATCH22, CATCH44, and CATCH66 spectra, carbonyl carbons and α -carbon peak areas match within 10% error suggesting minimal variation in cross-polarization. We believe our original observation of a decreasing ratio of cationic to anionic peptide still holds true and agrees with our prior observations of lysine residues exhibiting a weaker repulsion.

We added the following sentence, “1D ^{13}C spectra were collected using the composite-pulse multiCP method from Duan et al. which was validated with a series of similar co-assembled β -sheet peptide samples (Supplementary Fig. 8-13) and allows quantitative peak comparison.”

We updated the solid-state NMR methods section, “Quantitative ^1H - ^{13}C Cross-Polarization Magic Angle Spinning (CPMAS) measurements were conducted at 277K using the composite-pulse multiCP pulse sequence to compensate for effects of motion and relaxation. Samples were spun at 22 kHz to prevent spectral overlap from spinning sidebands. Measurements were run with 100 kHz decoupling, 0.2s ^1H recovery time, 2s recycle delay, and 14 $100\mu\text{s}$ CP periods to ensure equivalent cross-polarization.”

2. The absence of a "lag phase" in the ThT fluorescence build-up curves is remarkable. The authors should discuss this feature and how it may relate to the fibrillation mechanism.

We appreciate the reviewer's insight and agree that the lack of a lag phase is interesting. We have added text to the manuscript to briefly compare CATCH peptides to natural amyloid-like molecules such as amyloid- β and α -synuclein, for which kinetics have been extensively studied.

3. I would expect ionic strength to affect the kinetics and possibly the fibril morphologies observed for these peptides. The authors should test whether higher or lower ionic strengths have significant effects.

We appreciate this suggestion from the reviewer. We have included Thioflavin T (ThT) kinetic data at varying ionic strengths and composition to experimentally test this as shown in Supplemental Fig. 22-27. The results demonstrate that CATCH(4+) is more prone to self-association at the highest and lowest ionic strengths tested, through a mechanism that is currently unknown. In contrast, the self-association of the other CATCH peptides was not affected by ionic strength. We also found that the influence of ionic strength on co-assembly varies with the peptide pair. In particular, CATCH(4+/4-) co-assembly is fastest at intermediate ionic strength, whereas CATCH(6+/6-) co-assembly increased with ionic strength, such that co-assembly in 10x PBS was so rapid that time to half-max ThT signal could not be determined. We anticipate that ionic strength could affect peptide composition, and in turn, morphology of nanofibers. Such relationships would be difficult to deduce from TEM micrographs alone, so we have opted to pursue the effects of ionic strength on morphology in a later study. Text related to this has been added to the manuscript.

Reviewer #2

Overall, this de novo design of beta-structure-forming peptides is highly innovative and interesting, whose sequence, self-assembling structure, and ionic-binding activity of these peptides are also important for beta-sheet-forming diseases and self-assembling nanomaterials. The authors have presented significant computational data in both main text and Supplementary Information to demonstrate their designs. The paper can be published in its present form by including some discussion about how sequence-length affects the beta-sheet forming capacity and subsequent design, any suggestion?

We would like to thank the reviewer for their assessment of our work. We appreciate the suggestion to include some discussion on how sequence-length affects beta-sheet forming capacity and have included text within the discussion to address the comment and therefore, strengthen our manuscript.

Reviewer #3

This paper by Hudalla and colleagues is a fantastic addition to their very interesting body of work on "selective co-assembly" of peptides. The authors should be commended for the thorough nature of their approach, the nice blend of experimental and computational studies, and some creative molecular design. This paper will offer the peptide self-assembly community many new ideas in terms of facilitating systems specifically designed to co-assemble, and I look forward to the certain horizon for use of this approach to create bioactive nanostructures.

A few thoughts which arise from curiosity rather than criticism:

1. It is presumed the peptides are C-terminal amides, but this was not explicitly stated nor was the resin used listed in order to make this conclusion.

We apologize for this oversight. We have included the resin used in the peptide synthesis process to the Materials and Methods section to make this clearer.

2. Is the observation for a lack of self-association reliant on a symmetric nature of the peptides? For instance, would this general peptide approach with charge conserved work for a sequence where half the residues were D-amino acids (e.g., KQKFKFkfkqk) or the alternating structure was disrupted (e.g., KQKFKFGKFKQK)? Basically, is it just charge, or does the fit of the interface between the two molecules also play a role?

We appreciate these nested questions from the reviewer. In response to the question, ‘is it just charge?’, from our previously published work on CATCH peptides, we found that mutating the phenylalanine residues for prolines prevented the co-assembly of oppositely charged molecules (Seroski CMBE 2016, ref 52 in manuscript). These observations suggest that CATCH peptides do not co-assemble solely due to electrostatic charge, but rather that the fit of the interface between the molecules is important. However, it is unclear at this time if this is a universal property, or if instances exist in which peptides will co-assemble via electrostatic interactions alone. We recognize the importance of this consideration, and we have added some text related to our prior observations to the discussion in the manuscript.

The role of symmetry may be important and at present we have no data to support or refute the reviewer’s query regarding the importance of symmetry. The sequence space encompassing

peptides with a combination of L- and D-amino acids, as well as disrupted primary sequence, is vast, extending far beyond the scope of this report. Nonetheless, these are very interesting questions that will be worthwhile to pursue in future work, as the insights from such studies will likely lead to new opportunities to direct co-assembly processes.

3. The authors note a large role for electrostatic repulsion in preventing self-association in the single peptide systems. This is very reasonable. Given the studies were done in PBS, the Debye length is somewhere in the <1 nm range. One wonders whether electrostatic repulsion could be overcome if the Debye length could be further shortened by even higher osmolarity (e.g., 10x PBS).

We appreciate the reviewer's insight. To satisfy the reviewer's curiosity, we performed Thioflavin T (ThT) kinetic experiments on the single peptide systems (Supplemental Fig. 24-27). The data demonstrate that CATCH(4-), CATCH(6+), and CATCH(6-) do not self-associate even at higher ionic strengths (10x PBS); however, we found that CATCH(4+) exhibits higher ThT fluorescence in 10x PBS than in 1x or 5x PBS. This suggests that the electrostatic repulsion from the charged amino acid residues on CATCH(4+) can be overcome with high salt concentrations. This is consistent with unpublished work under review elsewhere, which includes zeta potential measurements suggesting that the CATCH(+) peptides have a lower surface charge than the CATCH(-) peptides. This likely reflects differences in the ionization potential of the side chains, as discussed in this manuscript.

REVIEWERS' COMMENTS:

Reviewer #1 (Remarks to the Author):

The authors have performed additional experiments and made additions to the text to address the points I raised in my review of the original version of this manuscript. All of my points have been fully addressed, and I have no additional comments. I thank the authors for taking the reviews seriously and addressing them thoroughly.

Reviewer #2 (Remarks to the Author):

The authors have addressed all reviewers' concerns, including mine, thus I recommend to publish this paper.

Reviewer #3 (Remarks to the Author):

I am satisfied with the author's responses to my questions and look forward to future work on this topic.